# Adolescent anxiety and pain problems: A joint, genome-wide investigation and pathway-based analysis

Sara Mascheretti[1,2], Diego Forni[3], Valentina Lampis[1,2], Luca Fumagalli[3], Stéphane Paquin[4], Till F. M. Andlauer[5], Wei Wang[6], Ginette Dionne[7], Mara R. Brendgen[8], Frank Vitaro[9,10], Isabelle Ouellet-Morin[11], Guy Rouleau[12], Jean-Philippe Gouin[13], Sylvana Côté[14], Richard E. Tremblay[15], Gustavo Turecki[16], Gabrielle Garon-Carrier[17], Michel Boivin[7]*, Marco Battaglia[18,19]*

1 Department of Brain and Behavioral Sciences, University of Pavia, Pavia, Italy, 2 Child Psychopathology Unit, Scientific Institute, IRCCS Eugenio Medea, Bosisio Parini, Italy, 3 Bioinformatics, Scientific Institute, IRCCS Eugenio Medea, Bosisio Parini, Italy, 4 Department of Psychology, The Pennsylvania State University, State College, PA, United States of America, 5 Department of Neurology, Klinikum rechts der Isar, School of Medicine, Technical University of Munich, Munich, Germany, 6 Centre for Complex Interventions Centre for Addiction and Mental Health, Toronto, Canada, 7 Ecole de Psychologie, Université Laval, Quebec City, QC, Canada, 8 Département de Psychologie, Universite du Quebec a Montreal, Montreal, QC, Canada, 9 Research Unit for Children's Psychosocial Maladjustment, Montreal, QC, Canada, 10 School of Psycho-Éducation, Université de Montréal, Québec City, QC, Canada, 11 School of Criminology, University of Montreal & Research Center of the Montreal Mental Health University Institute, Montreal, Canada, 12 Montreal Neurological Institute-Hospital, McGill University, Montreal, QC, Canada, 13 Department of Psychology, Concordia University, Montreal, Canada, 14 Département de Médecine Sociale et Préventive, Université de Montreal, Montreal, QC, Canada, 15 Départements de Pédiatrie et de Psychologie, Université de Montreal, Montreal, QC, Canada, 16 Douglas Research Centre, McGill University, Montreal, QC, Canada, 17 Department of Psychoéducation, Université de Sherbrooke, Québec, Canada, 18 Child, Youth and Emerging Adults Programme Centre for Addiction and Mental Health, Toronto, Canada, 19 Department of Psychiatry, University of Toronto, Toronto, Canada

* marco.battaglia@camh.ca (MB); michel.boivin@psy.ulaval.ca (MB)

**Data Availability Statement:** Data cannot be shared publicly due to ethical restrictions implied by participants' informed consent, but data requests may be submitted to the Research Unit

## Abstract

Both common pain and anxiety problems are widespread, debilitating and often begin in childhood-adolescence. Twin studies indicate that this co-occurrence is likely due to shared elements of risk, rather than reciprocal causation. A joint genome-wide investigation and pathway/network-based analysis of adolescent anxiety and pain problems can identify genetic pathways that subserve shared etiopathogenetic mechanisms. Pathway-based analyses were performed in the independent samples of: The Quebec Newborn Twin Study (QNTS; 246 twin pairs and 321 parents), the Longitudinal Study of Child Development in Quebec (QLSCD; n = 754), and in the combined QNTS and QLSCD sample. Multiple suggestive associations ($p < 1 \times 10^{-5}$), and several enriched pathways were found after FDR correction for both phenotypes in the QNTS; many nominally-significant enriched pathways overlapped between pain problems and anxiety symptoms (uncorrected $p < 0.05$) and yielded results consistent with previous studies of pain or anxiety. The QLSCD and the combined QNTS and QLSCD sample yielded similar findings. We replicated an association between the pathway involved in the regulation of myotube differentiation (GO:0010830) and both pain and anxiety problems in the QLSDC and the combined QNTS and QLSCD sample. Although limited by sample size and thus power, these data provide an initial

on Children's Psychosocial Maladjustment Website (http://www.gripinfo.ca/grip/public/www/etudes/en/dadprocedures.asp) and to the Institut de la Statistique du Québec (https://www.jesuisjeserai.stat.gouv.qc.ca/informations_chercheurs/acces_an.html).

**Funding:** Both studies were supported by grants from the Fonds de recherche du Quebec – Societe et Culture (FRQ-SC), Fonds de recherche du Quebec – Sante (FRQ-S), the Réseau québecois sur le suicide, les troubles de l'humeur et les troubles associés, the Social Science and Humanities Research Council of Canada (SSHRC), the Canadian Institutes for Health Research (CIHR), and Ste. Justine Hospital's Research Center. The QNTS was also supported by funding from the National Health Research Development Program, Université Laval, and Université de Montreal. The QLSCD was also supported by funding from the Gouvernement du Québec, the Lucie and André Chagnon Foundation, the Robert-Sauvé Research Institute of Health and Security at Work, and the Institut de la statistique du Quebec. Dr Battaglia's research is supported by the Canadian Institutes of Health Research, the Natural Sciences and Engineering Research Council of Canada, the Quebec Pain Research Network, the Cundill Foundation and the CAMH Foundation; he has been supported by a research grant from the Université Laval Merck Sharpe Dome Foundation, has received stipends from the Canadian Psychiatric Association and from Servier International Medical Publishing Division. Dr. Boivin (Tier1), G. Rouleau (Tier 1), G. Turecki (Tier 1), I. Ouellet-Morin (Tier 2), J.-P. Gouin (Tier 2), and G. Garon-Carrier (Tier 2) are supported by the Canada Research Chair Program. Dr. Mascheretti was supported by Italian Ministry of Health Grants (Ricerca Corrente 2021). Dr. Fumagalli was supported by "5 per mille" funds for biomedical research.

**Competing interests:** The authors have declared that no competing interests exist.

support to conjoint molecular investigations of adolescent pain and anxiety problems. Understanding the etiology underlying pain and anxiety co-occurrence in this age range is relevant to address the nature of comorbidity and its developmental pathways, and shape intervention. The replication across samples implies that these effects are reliable and possess external validity.

## Introduction

Between 8% and 12% of adolescents aged 11–17 years and 16%-20% of youths or young adults in North-America and Europe suffer from chronic pain [1, 2]. Adolescent pain mostly occurs without a recognizable association with a medical condition [2] and, similarly to adult pain [3], it is often persistent [1, 2, 4] and is frequently associated with anxiety and depressive symptoms [5, 6].

A deep understanding of the nature of the co-occurrence between common and persistent pain and anxiety/depression (internalizing conditions) is necessary to inform diagnostic reasoning, guide clinical practice, develop new medications, and help reducing opioid prescriptions and abuse [4, 7].

The majority of adult twin studies reveal that the co-occurrence of chronic pain and internalizing conditions is best explained by the substantial covariation of genetic and environmental factors, with only a minority of twin studies supporting the likelihood of phenotypic causation [8]. Similarly, the only two adolescent pain twin studies (one of which is longitudinal) [4] indicate that the co-occurrence of anxiety and adolescent pain problems is accounted for by genetic and environmental factors [4, 9] that are shared by the two phenotypes and has a similar etiology for boys and girls [4], even though both anxiety and pain problems are more prevalent among girls.

The molecular genetic bases of anxiety and pain have been investigated separately through GWAS and polygenic risk scores studies in adults [e.g., 10, 11], and less commonly in adolescents [e.g., 12], but no molecular genetic study investigated the shared etiopathogenetic mechanisms underlying both pain and anxiety in adolescents. Supporting this endeavor, a strong genetic correlation was found between internalizing symptoms and migraine [13]. Furthermore, understanding the etiology that underlies the adolescent pain and anxiety co-occurrence can help address fundamental questions such as the nature and the developmental pathways of comorbidity, and shape intervention [6]. Without this knowledge, the costs related to pain and its treatment cannot be reduced [14–16], or a culture of risk identification be set into practice.

Here, we present the results of an exploratory joint genome-wide investigation and pathway/network-based analyses of adolescent pain and anxiety problems in two independent longitudinal cohorts of Canadian adolescents, i.e. the Quebec Newborn Twin Study (QNTS) and the Longitudinal Study of Child Development in Quebec (QLSCD), as well as in the combined QNTS and QLSCD samples.

## Materials and methods

### Sample

**Quebec Newborn Twin Study (QNTS).** Our discovery cohort is part of the Quebec Newborn Twin Study (QNTS), an ongoing prospective longitudinal follow-up of a birth cohort of twins born between 1995 and 1998 in the Montreal area [17]. A population-based sample of 662 families of twins were initially enrolled in QNTS, and then assessed longitudinally at

regular intervals. Zygosity was initially assessed via questionnaire [18] and confirmed with DNA tests on a subsample of same-sex pairs showing a 96% correspondence [19, 20]. The latest completed QNTS wave of assessment (age 19) yielded a cumulative attrition rate of 23% since the study's inception, with 73% of remaining participants in complete pairs. The QNTS encompasses a broad range of physiological, cognitive, behavioral, school-, and health-related phenotypes collected prospectively and repeatedly with multi-informant and multimethod measures. For the purpose of this study, we included participants from 501 twin pairs with assessments of pain problems and anxiety symptoms at age 12, 13, and 14 years [4]. All procedures performed in this study were in accordance with the ethical standards of Université Laval and Sainte-Justine Hospital and with the 1964 Helsinki declaration and its later amendments or comparable ethical standards. Informed written consent was obtained by parents of all the participants included in the study.

**Longitudinal Study of Child Development in Quebec (QLSCD).** Our replication cohort is part of the QLSCD, a population-based representative sample of children born in the Canadian Province of Quebec between October 1997 and July 1998 [21]. This cohort includes assessments drawn from the National Longitudinal Survey of Children and Youth in Canada, initiated in 1994–95 (NLSCY: https://crdcn.org/datasets/nlscy-national-longitudinal-survey-children-andyouth). The QLSCD included extensive assessments of early cognitive, emotional, and behavioral problems, as well as a wider range of measurements of the quality of childcare environments and children's behavior. From an initial pool of 2,940 families, 2,120 gave their consent to participate in subsequent data collections, and constitute the longitudinal QLSCD sample [21]. The present study includes 1,437 QLSCD individuals who provided responses to pain problems and anxiety symptoms assessments across ages 12, 13, and 14. The ethics review committee of the Québec Institute of Statistics, which was responsible for data collection, approved this study. Before participating in the study, all families received detailed information by mail on the study aims and procedures, and all signed an informed written consent form at each measurement time.

## Phenotypes

**Quebec Newborn Twin Study (QNTS).** At age 12, 13, and 14, twins were asked about 6 common pain problems via 6 questions embedded in a general health questionnaire. These probed the frequency of: headaches, back pains, abdominal pain, chest pains, stabbing/throbbing pain, gastric pain/nausea, over the past academic year, on a 4-point scale ranging from 0 (never) to 3 (often). To obtain a concise measure of the frequency of pain problems, ratings were summed at each assessment time (3 ratings) [4]. Moreover, at age 12, 13, and 14, twins rated 7 items from the Children's Manifest Anxiety Scale (e.g., During the past month: Did you worry about what was going to happen?) on a 4-point scale ranging from 1 (never) to 4 (very often) [22]. Across the longitudinal assessment waves, the Cronbach's alpha indices of the pain items' cumulative scores (range: 0.71–0.74) and the anxiety symptoms mean scores (0.82–0.84) were acceptable [4]. Because of the moderate-to-strong covariance of pain problems and anxiety symptoms across the 12, 13, and 14 assessment waves at age, suggesting temporal stability [4], we averaged these scores to obtain: a QNTS_Mean Pain, and a QNTS_Mean Anxiety index for each twin (S1 Table).

**Longitudinal Study of Child Development in Quebec (QLSCD).** At age 12 and 13, each subject rated presence of headaches on a dichotomous (Yes/No) scale and 3 items for anxiety symptoms (i.e., "too fearful or anxious", "worried" and "nervous, high-strung or tense") on a 3-point scale ranging from 0 (never occurs) to 2 (frequently occurs). Average anxiety scores were then computed at each assessment time, with Cronbach alphas averaging 0.80 across measurements [23].

Regarding pain problems, QLSCD missing values were then imputed to increase power and reduce potential bias incurred by uneven missing. The correlation structure presented by the available QNTS measures was used as the basis of imputation and sex, average anxiety scores and headaches at ages 12 and 13 were used as anchor variables. After harmonizing the common measures between QNTS and QLSCD and evaluating their predictability, we imputed the missing values by using predictive mean matching [24] and imputed 70 datasets to attain a 99% efficiency. We then created two cumulative scores (i.e., QLSCD_Mean Pain and QLSCD_Mean Anxiety) by estimating the median value across the 70 imputed datasets within pain problems and anxiety symptoms (S1 Table).

## Genotyping, quality control (QC) and imputation

**Quebec Newborn Twin Study (QNTS).** Blood and saliva samples were collected from the twins and their parents for a subsample of QNTS families (909 twins -including 174 MZ twins- and 407 parents) [17]. For MZ twin pairs, one child per pair based on the maximum availability of phenotypic data, or otherwise randomly, was selected [17]. DNA was extracted from blood or saliva samples and prepared for genotyping using standard protocols. Genome-wide genotype data were generated using semi-custom chip based on Illumina® PsychArray-24v1.1 Beadchip which is a cost-effective, high-density microarray developed in collaboration with the Psychiatric Genomics Consortium for large-scale genetic studies focused on psychiatric predisposition and risk. The custom chip had an additional 790 SNPs selected based on prior knowledge that suggested the presence of association with psychological phenotypes and social behaviors. Data was processed using Illumina's GenomeStudio platform, following the manufacturer's guidelines. All the collected samples passed a first round of QC [25]. In this first round, SNPs with call rates <98% or a minor allele frequency (MAF) <1% were removed as well as samples with <98% genotyping rate. In addition, samples were excluded for the following reasons: sex mismatch, genetic duplicates or cryptic relatives (pi-hat≥12.5), deviation from the mean autosomal heterozygosity (>4 SD). Multi-dimensional scaling (MDS) analysis of genome-wide genotype data was also used to identify any subjects that did not cluster together with the majority of the dataset, and these were discarded, as were any outliers for genome-wide heterozygosity. Furthermore, individuals were excluded based on inconsistent relationship status. Genotype phasing was conducted using SHAPEIT v2 (r837) [26]. Imputation of variants was carried out using IMPUTE2 v2.3.2 [27] and the 1000 Genomes Phase 3 reference panel. We then converted probabilities to best-guess genotypes using PLINK with the default 0.1 threshold. After imputation, we removed variants with a MAF <1%, an HWE test $p<1\times10^{-6}$, and an INFO metric <0.8. At the end of this QC and imputation process, we had data for 4,385,954 polymorphisms and 246 twin pairs with complete phenotypic information (encompassing 352 twins: 109 MZ and 243 DZ) and 321 parents. Descriptive statistics of the phenotypic distributions are reported in S1 Table. Post-hoc power calculations were conducted using the Genetic Power Calculator [28] to estimate the smallest effect size that our samples could detect with 80% statistical power. The analysis was modelled for an allele frequency of 0.05. No dominance effects and perfect linkage disequilibrium were assumed; alpha was set at $5^{10-8}$. Under these assumptions, the minimal effect sizes predicted to be detectable with 80% power was 0.08% in our twin pairs' cohort (n = 246).

**Longitudinal Study of Child Development in Quebec (QLSCD).** Genotypes were also determined using the Illumina Infinium PsychArray-24v1.1 Beadchip which, after following the same quality control and imputation steps as described for the QNTS, yielded 5,129,426 genetic variants (4,028,157 polymorphisms shared with the QNTS) for 754 participants with complete phenotypic and genotypic data. Descriptive statistics of the phenotypic distributions

are reported in S1 Table. As in the QNTS, post-hoc power calculations were conducted using the Genetic Power Calculator [28] to estimate the smallest effect size that our samples could detect with 80% statistical power. The analysis was modelled for an allele frequency of 0.05. No dominance effects and perfect linkage disequilibrium were assumed; alpha was set at $5^{10-8}$. Under these assumptions, the minimal effect sizes predicted to be detectable with 80% power was 0.05% in our sample (n = 754).

**Pathway/Network-based association analysis.** Although the phenotypic distributions were skewed (S1 Table), there is ample evidence that in behavioral phenotypes a continuous distribution can be safely assumed [e.g., 29]. Genome-wide association analyses were conducted using mean pain and mean anxiety scores separately within each dataset. Multiple analyses of sex effects among QNTS participants showed that models that imposed sex-related constraints on anxiety or pain did not yield better fit than models that imposed no such constraints [4, 30]. Within the limitation of this quantitative genetic approach, this indicates a similar etiology for boys and girls alike for both anxiety and pain, and therefore participants of both sexes were pooled together in the analyses. In the QNTS cohort, the 'total' association option of the PLINK v1.90's QFAM function (which tests for association at each SNP by regressing trait scores on genotypes with a special permutation test correcting for family structure and controlling for non-normally distributed phenotypes) was used (http://pngu.mgh.harvard.edu/~purcell/plink/) [31]. In the QLSCD sample, linear regression was conducted in PLINK. To assess significance levels empirically, an adaptive permutation procedure (—aperm) which allows to relax assumptions about normality of continuous phenotypes, was used with the following parameters: minimum number of permutations per SNP = 1,000, maximum number of permutations per SNP = 1,000,000,000, alpha (determining the threshold for pruning *p-values*) = 0, beta (determining the width of confidence interval on empirical p-value) = 0.01, initial interval (number of permutations) to prune SNP test list = 100, and rate of increase of the initial interval to prune SNP test list = 0.001 [32].

A pathway/network-based enrichment analysis was run using the PLINK results with QNTS_Mean Pain, QNTS_Mean Anxiety, QLSCD_Mean Pain, and QLSCD_Mean Anxiety as traits. The top variants per associated locus were determined using clumping of the summary statistics in PLINK implementing the following arguments:—clump-p1 0.001 (*p-value* threshold for index SNPs),—clump-p2 0.01 (*p-value* threshold for clumped SNPs),—clump-r2 0.5 (LD ($R^2$) threshold for clumping),—clump-kb 250 (kilobase threshold for clumping). The generated non-overlapping genomic regions were used as input in the Broad-Enrich tool [33]. This method tests broad genomic regions for enriched biological pathways or Gene Ontology (GO) terms. More specifically, it scores gene loci according to the proportion of the locus covered by all predefined regions overlapping it (coverage proportion). Broad-Enrich then uses a logistic regression to test for association between this proportion and gene set membership. The model also adjusts for any bias in gene locus coverage relative to locus length [33]. Gene loci were defined as the start and end of each human GRCh37 gene, plus upstream and downstream 10 kilobases. Direct association between 16,496 unique genes and their corresponding GeneOntology (GO) Biological Process (n = 12,143) were used in the analysis. Broad-Enrich was run in the R environment v4.0.4, with dedicated packages [33].

## Results

### Quebec Newborn Twin Study (QNTS)

To test genome-wide association, we ran the 'total' association option of the PLINK v1.90's QFAM function (see the 'Pathway/network-based association analysis' paragraph). S1 Fig shows genome-wide Manhattan Plots for both QNTS_Mean Pain and QNTS_Mean Anxiety.

While no genome-wide significant association was observed, several suggestive associations (permuted $p<1\times10^{-5}$; S2A and S2B Table) were found for both phenotypes.

We then assessed evidence for an excess of association signals within the genes of the same biological process for both phenotypes. Several enriched pathways with nominally-significant uncorrected $p<0.05$ were found for both QNTS_Mean Pain and QNTS_Mean Anxiety (S3A and S3B Table). Among these pathways, after FDR correction (5%), 6 and 13 pathways, respectively, were significant for QNTS_Mean Pain (S3A Table) and QNTS_Mean Anxiety (S3B Table). Although no GOcategories corrected for multiple comparisons were shared between QNTS_Mean Pain and QNTS_Mean Anxiety, 37 pathways with uncorrected $p<0.05$ were overlapping between pain problems and anxiety symptoms (S4 Table).

## Longitudinal Study of Child Development in Quebec (QLSCD)

To test genome-wide association, linear regression with an adaptive permutation procedure was conducted in PLINK (see the 'Pathway/network-based association analysis' paragraph). S2 Fig shows genome-wide Manhattan Plots for both QLSDC_Mean Pain and QLSDC_Mean Anxiety. Again, some suggestive associations were observed ($p<1\times10^{-5}$; S5A and S5B Table), although they did not reach genome-wide significance.

We then assessed evidence for an excess of association signals within the genes of the same biological pathways on both pain problems and anxiety symptoms. Several enriched pathways with nominally-significant uncorrected $p<0.05$ were found for both QLSDC_Mean Pain and QLSDC_Mean Anxiety (S6A and S6B Table). Among these pathways, after FDR correction, 3 and 5 pathways, respectively, were significant for QLSDC_Mean Pain (S6A Table) and QLSDC_Mean Anxiety (S6B Table). Although no GO categories corrected for multiple comparisons were shared between QLSDC_Mean Pain and QLSDC_Mean Anxiety, 107 pathways with uncorrected $p<0.05$ overlapped pain problems and anxiety symptoms (S7 Table). The overlapping pathways found nominally significant in the QNTS were tested for replication in QLSDC. We found that the pathway involved in the regulation of myotube differentiation (GO:0010830; Fig 1) overlapped pain problems and anxiety symptoms in both the QNTS and the QLSDC (S7 Table).

This figure depicts the interaction network of genes associated with the GO category GO:0010830 (black circles) and their connecting genes (gray circles), which associated with pain problems and anxiety symptoms in the QNTS and replicated in the QLSDC. Node size represents the connection degree among genes. Interaction types are shown as colored edges and are reported in the legend. This network was visualized using the GeneMANIA Cytoscape plugin (http://www.genemania.org/plugin/) which identifies the most related genes to a query gene set using a guilt-by-association approach. The plugin uses a large database of functional interaction networks from multiple organisms and each related gene is traceable to the source network used to make the prediction.

## Combined QNTS and QLSCD sample

As 4,028,157 polymorphisms were shared and the phenotypic distribution did not differ between the two cohorts (Mann-Whitney U test = 132,241.000, SE = 4,945.508, $p = 0.925$ and Mann-Whitney U test = 128,134.500, SE = 4,946.462, $p = 0.356$ for pain problems and anxiety symptoms, respectively), we merged the genomic regions for which we found nominally significant results in both the QNTS and QLSCD to achieve stronger statistical power and thus test the robustness of the previous findings. This led to a single region list containing the genomic regions surrounding the top variants of both QNTS and QLSCD samples. Several enriched pathways with uncorrected $p<0.05$ were found for Mean Pain and Mean Anxiety (S8A and S8B Table). Among these pathways, and after FDR correction, 4 pathways stayed significant

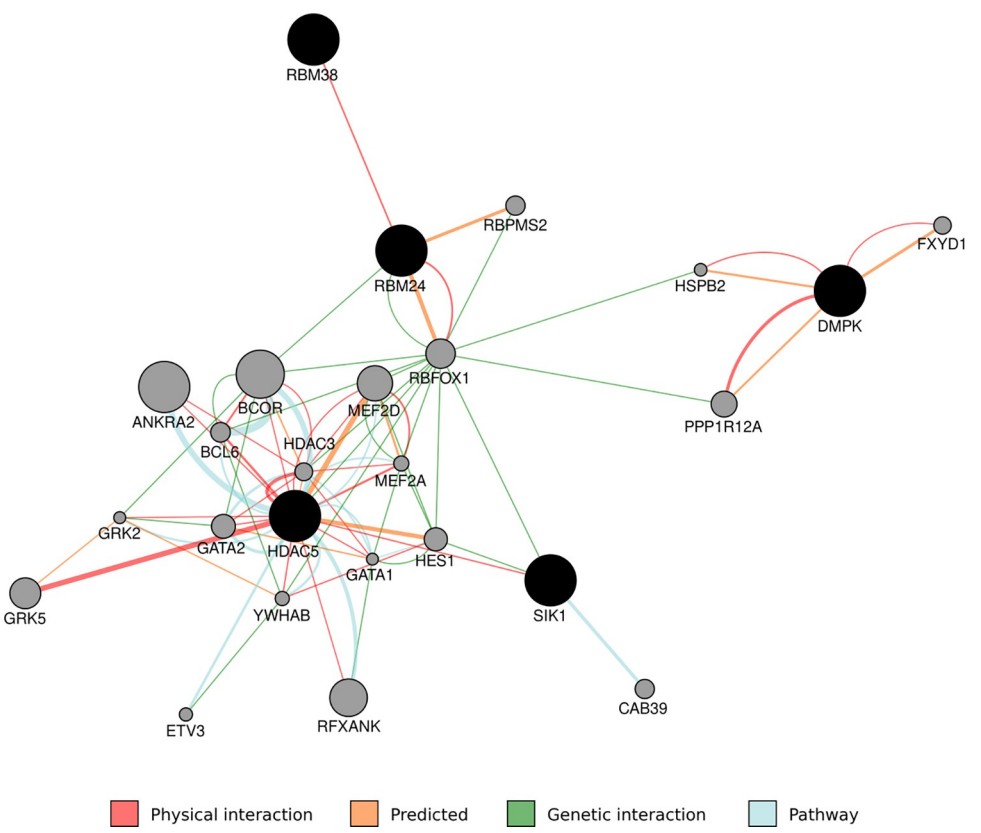

**Fig 1. Regulation of myotube differentiation network (GO:0010830).**

for Mean Pain (S8A Table), but none survived for Mean Anxiety (S8B Table). Although no GO categories corrected for multiple comparisons were shared between Mean Pain and Mean Anxiety, 76 pathways with uncorrected $p<0.05$ overlapped pain problems and anxiety symptoms (S9 Table). The overlapping pathway found nominally significant in the QNTS and then replicated in the QLSDC cohort (i.e., GO:0010830), was again tested in the combined QNTS and QLSCD sample. Similar to what we observed in the QNTS and QLSDC independent cohorts, a nominally significant association emerged for the pathway involved in the regulation of myotube differentiation in both pain problems and anxiety symptoms in the combined analysis (S9 Table).

## Discussion

This exploratory study aimed to identify genetic pathways associated with shared etiopathogenetic mechanisms underlying the co-occurrence of pain and anxiety problems. We conducted a joint genome-wide investigation and pathway/network-based analysis in two independent longitudinal cohorts of Canadian adolescents, the QNTS and the QLSCD, and then, in the combined samples. In particular, we sought to detect an excess of association signals within genes associated with various GO biological processes.

Overall, the pathway involved in the regulation of myotube differentiation (GO:0010830) was found to be associated with pain problems and anxiety symptoms in the QNTS (S4 Table). This finding was replicated in the QLSDC (S7 Table). Testing for association with independent discovery and replication samples is a well-known approach in molecular genetics, and was adopted here to corroborate results from the discovery sample and minimize type I error [e.g.,

34]. A similar pattern of results was found in the combined QNTS and QLSDC sample (S9 Table). Among the significant candidate genes within this nominally-significant enriched pathway, some have been previously associated with pain problems and anxiety symptoms. The *DMPK* gene encodes a serine-threonine kinase protein whose substrates include myogenin, the beta-subunit of the L-type calcium channels, and phospholemman. The *HDAC5* gene is characterized by histone deacetylase activity, and represses transcription when tethered to a promoter. The nociceptive hypersensitivity of this gene has been indicated in different pain models [35], and its expression in the hippocampus plays a pivotal role in susceptibility/ resilience to chronic stress [36] and is altered by epigenetic processes linked to an interaction between isolated housing and social stress [37]. Finally, the *HDCA5* gene is also involved in the ketamine-induced transcriptional regulation of the brain derived neurotrophic factor (*BDNF*), and its phosphorylation regulates the therapeutic actions of ketamine—a therapeutic agent for the treatment of depression and anxiety [38].

Other potentially interesting results supporting the presence of shared etiopathogenetic mechanisms underlying the phenotypic co-occurrence of pain and anxiety problems emerged in the QNTS, the QLSCD or the combined sample, although they need replications in other independent datasets. Several enriched pathways with uncorrected $p < 0.05$ were overlapping between pain problems and anxiety symptoms in the QNTS (S4 Table), the QLSCD (S7 Table), or the combined sample (S9 Table). Within these nominally significant overlapping pathways, some have been previously associated with pain problems and/or anxiety symptoms (e.g., the brain derived neurotrophic factor signaling pathway (GO:0031547) [39, 40]; the glucocorticoid receptor signaling pathway (GO:0042921) [41, 42]. Moreover, some significant genes within these nominally significant overlapping pathways have been previously associated with pain problems and/or anxiety symptoms (e.g., the *ZNRD1* gene (S4 Table) [43]; the *MDK* gene (S4 and S9 Tables) [44]; the *FOXO3* gene (S4 Table) [45]; the *COMT* gene (S9 Table) [46, 47]; the *NFATC4* gene (S7 and S9 Tables) [40, 48]). Interestingly, after FDR correction, several enriched pathways reached significance for pain problems and/or anxiety symptoms in the QNTS (S3A and S3B Table), the QLSCD (S6A and S6B Table), and the combined sample (S8A and S8B Table). Among these empirically significant enriched biological processes pathways, the glucocorticoid receptor signaling pathway (GO:0042921; S6A, S6B, S8A and S8B Tables) has been shown to be associated with common behavioral and/or somatic complex disorders, such as anxiety, chronic pain and fatigue syndromes, and to contribute to the development of neuropathic pain [41, 42]. Moreover, some significant genes within these empirically significant enriched pathways have been previously associated with pain problems (e.g., the *PTGS2* gene (S3A and S8A Tables) [49]; the *HLA-A* gene (S3A and S8A Tables) [50]) and anxiety symptoms (e.g., the *SYNGAP1* gene (S3B and S8B Tables) [51]; the *ULK4* gene (S3B and S8B Tables) [52]).

The current findings are consistent with previous evidence showing a significant genetic overlap between migraine and major depressive disorder [53] and a substantial genetic similarity between individuals with comorbid depression and migraine and individuals with depression alone [54]. Moreover, the current findings integrate and expand actual knowledge by showing that the observed phenotypic co-occurrence between pain problems and anxiety symptoms is accounted for by a substantial overlap of genetic factors [8]. By testing the genetic association with average scores of assessments collected through early adolescence, our results agreed with previous multivariate longitudinal twin studies demonstrating that genetic factors that are common to both phenotypes support the temporal stability between pain problems and anxiety symptoms [4, 8]. We can therefore hypothesize that variation at the above-described pathways accounts for function at some intermediate phenotypes, which would act as latent factors underlying the co-occurrence between common and persistent pain and anxiety (internalising conditions). Quite compelling is the possible interplay of the biological

systems of harm detection, such as the perception of pain or the detection of dyspnoea (a key mechanism of panic anxiety), in emotion-related brain networks [54, 55]. This suggests that anxiety and pain may be linked together under the umbrella of altered interoception, an emerging field of investigation at the crossroads of mental and physical health [55, 56]. Ultimately, these findings may help healthcare practitioners and policymakers better inform diagnostic reasoning, develop more effective standardized measurement tools as well as treatment interventions for pain and anxiety/depression (internalising conditions), and reduce the social and economic burden of these conditions.

The findings of this study need to be interpreted in the context of several limitations. First, although this is the first exploratory molecular genetic study investigating the etiopathogenetic mechanisms subserving pain and anxiety in two independent, well-characterized longitudinal cohorts, the sample size was small for genome-wide studies. Moreover, small sample size did not allow us to stratify the analyses by sex. Although available twin adolescent data indicate a similar etiology of pain and anxiety for boys and girls [4, 9, 30], these remain quantitative genetic modeling applications, so that molecular genetic replications in larger datasets are needed. Second, a pre-spotted microarray containing candidate genes for psychiatric predisposition and risk has been genotyped. This may have limited the potential of the study since identification of pain-relevant candidate genes is dependent on the genes with overlapping phenotypes and therefore skews the observations. Third, as in most large-scale epidemiological studies of adolescent pain, self-report measures of anxiety and pain and the reliance on single informant data, were available. Fourth, our questionnaires lacked items addressing muscular-relevant observations. Fifth, although the QNTS and the QLSCD shared a high percentage of best-guess genotyped polymorphisms (about 80%), this led to a different number of genotyped polymorphisms between the two samples. Sixth, we average pain and anxiety items in the 12–14 years' age range into unitary mean scores of general pain and anxiety proclivity. Consequently, we cannot delineate whether temporal stability and longitudinal correlation between phenotypes is attributable to stable genetic influences over time, and we cannot apply our findings to specific pain types at particular body sites or to specific anxiety symptoms.

## Conclusions

In addressing the biological bases of the co-occurrence between adolescent pain and anxiety, we aimed at better characterizing the nature of their comorbidity, their shared developmental pathways, and the bases for sounder intervention [6]. The lack of data, and/or interest for these questions have already negatively impacted on the societal costs of pain. Pain problems not caused by identified medical conditions begins early in life, are common among adolescents [1, 4, 8, 30, 57], are associated with internalizing symptoms [4, 6, 58, 59], persist into youth and adulthood [1, 57, 58, 60], and are associated with negative long-term outcomes [61]. Adolescent anxiety and persistent pain likely constitute a gateway to premature, more prolonged, and more hazardous opioid prescription [14–16]. Inasmuch as a culture of risk identification has not yet entered into practice, and specific treatments for young people who present with both pain and anxiety are lacking, this type of research can prove valuable. By showing potential shared genomic risk between pain problems and anxiety symptoms, these findings can inform diagnostic reasoning, guide clinical practice, and support the development of new medications.

## Supporting information

**S1 Table. Descriptive statistics of the phenotype's distribution in the QNTS and QLSCD cohorts.**
(DOCX)

**S2 Table. Top associations (*p-value* < 1x10^{-5}) of the genome-wide analysis (QNTS_Mean Pain in a and QNTS_Mean Anxiety in b).**
(DOCX)

**S3 Table. Results of the pathway-based analysis of QNTS_Mean Pain (a) and QNTS_Mean Anxiety (b) (uncorrected *p-value* < 0.05).**
(DOCX)

**S4 Table. Overlapping enriched pathways between QNTS_Mean Pain and QNTS_Mean Anxiety (uncorrected p-value < 0.05).**
(DOCX)

**S5 Table. Top associations (*p-value* < 1x10^{-5}) of the genome-wide analysis (QLSDC_Mean Pain in a and QLSDC_Mean Anxiety in b).**
(DOCX)

**S6 Table. Results of the pathway-based analysis of QLSCD_Mean Pain (a) and QLSCD_Mean Anxiety (b) (uncorrected *p-value* < 0.05).**
(DOCX)

**S7 Table. Overlapping enriched pathways between QLSCD_Mean Pain and QLSCD_Mean Anxiety (uncorrected p-value < 0.05).**
(DOCX)

**S8 Table. Results of the pathway-based analysis of Mean Pain (a) and Mean Anxiety (b) (uncorrected p-value < 0.05).**
(DOCX)

**S9 Table. Overlapping enriched pathways between Mean Pain and Mean Anxiety (uncorrected p-value < 0.05).**
(DOCX)

**S1 Fig. Genome-wide Manhattan Plots for QNTS_Mean Pain and QNTS_Mean Anxiety.**
(TIF)

**S2 Fig. Genome-wide Manhattan Plots for QLSDC_Mean Pain and QLSDC_Mean Anxiety.**
(TIF)

## Acknowledgments

We are grateful to all families and participants who took part in the study. We thank the GRIP staff for data collection and management.

## Author Contributions

**Conceptualization:** Sara Mascheretti, Michel Boivin, Marco Battaglia.

**Data curation:** Sara Mascheretti, Diego Forni, Stéphane Paquin, Till F. M. Andlauer.

**Formal analysis:** Sara Mascheretti, Diego Forni, Valentina Lampis, Luca Fumagalli, Wei Wang.

**Funding acquisition:** Ginette Dionne, Mara R. Brendgen, Frank Vitaro, Michel Boivin.

**Methodology:** Sara Mascheretti, Diego Forni.

**Supervision:** Michel Boivin, Marco Battaglia.

**Writing – original draft:** Sara Mascheretti, Diego Forni, Valentina Lampis.

**Writing – review & editing:** Luca Fumagalli, Stéphane Paquin, Till F. M. Andlauer, Wei Wang, Ginette Dionne, Mara R. Brendgen, Frank Vitaro, Isabelle Ouellet-Morin, Guy Rouleau, Jean-Philippe Gouin, Sylvana Côté, Richard E. Tremblay, Gustavo Turecki, Gabrielle Garon-Carrier, Michel Boivin, Marco Battaglia.

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
