## [Decision Letter · Decision Letter 0]

13 Mar 2023

PONE-D-22-19260Adolescent anxiety and pain problems: a joint, genome-wide investigation and pathway-based analysisPLOS ONE

Dear Dr. Battaglia,

Thank you for submitting your manuscript to PLOS ONE. After careful consideration, we feel that it has merit but does not fully meet PLOS ONE’s publication criteria as it currently stands. Therefore, we invite you to submit a revised version of the manuscript that addresses the points raised during the review process.

We look forward to receiving your revised manuscript.

Kind regards,

Toryn Poolman

Academic Editor

PLOS ONE

Journal Requirements:

3. PLOS requires an ORCID iD for the corresponding author in Editorial Manager on papers submitted after December 6th, 2016. Please ensure that you have an ORCID iD and that it is validated in Editorial Manager. To do this, go to ‘Update my Information’ (in the upper left-hand corner of the main menu), and click on the Fetch/Validate link next to the ORCID field. This will take you to the ORCID site and allow you to create a new iD or authenticate a pre-existing iD in Editorial Manager. Please see the following video for instructions on linking an ORCID iD to your Editorial Manager account: https://www.youtube.com/watch?v=_xcclfuvtxQ.

“We are grateful to all families and participants who took part in the study. We thank the GRIP staff for data collection and management. Both studies were supported by grants from the Fonds de recherche du Quebec – Societe et Culture (FRQ-SC), Fonds de recherche du Quebec – Sante (FRQ-S), the Réseau québecois sur le suicide, les troubles de l’humeur et les troubles associés, the Social Science and Humanities Research Council of Canada (SSHRC), the Canadian Institutes for Health Research (CIHR), and Ste. Justine Hospital’s Research Center. The QNTS was also supported by funding from the National Health Research Development Program, Université Laval, and Université de Montreal. The QLSCD was also supported by funding from the Gouvernement du Québec, the Lucie and André Chagnon Foundation, the Robert-Sauvé Research Institute of Health and Security at Work, and the Institut de la statistique du Quebec. Dr Battaglia's research is supported by the Canadian Institutes of Health Research, the Natural Sciences and Engineering Research Council of Canada, the Quebec Pain Research Network, the Cundill Foundation and the CAMH Foundation; he has been supported by a research grant from the Université Laval Merck Sharpe Dome Foundation, has received stipends from the Canadian Psychiatric Association and from Servier International Medical Publishing Division. Dr. Boivin (Tier1), G. Rouleau (Tier 1), G. Turecki (Tier 1), I. Ouellet-Morin (Tier 2), J.-P. Gouin (Tier 2), and G. Garon-Carrier (Tier 2) are supported by the Canada Research Chair Program. Dr. Mascheretti was supported by Italian Ministry of Health Grants (Ricerca Corrente 2021). Dr. Fumagalli was supported by "5 per mille" funds for biomedical research”

“We are grateful to all families and participants who took part in the study. We thank the GRIP staff for data collection and management. Both studies were supported by grants from the Fonds de recherche du Quebec – Societe et Culture (FRQ-SC), Fonds de recherche du Quebec – Sante (FRQ-S), the Réseau québecois sur le suicide, les troubles de l’humeur et les troubles associés, the Social Science and Humanities Research Council of Canada (SSHRC), the Canadian Institutes for Health Research (CIHR), and Ste. Justine Hospital’s Research Center. The QNTS was also supported by funding from the National Health Research Development Program, Université Laval, and Université de Montreal. The QLSCD was also supported by funding from the Gouvernement du Québec, the Lucie and André Chagnon Foundation, the Robert-Sauvé Research Institute of Health and Security at Work, and the Institut de la statistique du Quebec. Dr Battaglia's research is supported by the Canadian Institutes of Health Research, the Natural Sciences and Engineering Research Council of Canada, the Quebec Pain Research Network, the Cundill Foundation and the CAMH Foundation; he has been supported by a research grant from the Université Laval Merck Sharpe Dome Foundation, has received stipends from the Canadian Psychiatric Association and from Servier International Medical Publishing Division. Dr. Boivin (Tier1), G. Rouleau (Tier 1), G. Turecki (Tier 1), I. Ouellet-Morin (Tier 2), J.-P. Gouin (Tier 2), and G. Garon-Carrier (Tier 2) are supported by the Canada Research Chair Program. Dr. Mascheretti was supported by Italian Ministry of Health Grants (Ricerca Corrente 2021). Dr. Fumagalli was supported by "5 per mille" funds for biomedical research.”

Reviewers' comments:

Reviewer's Responses to Questions

**Comments to the Author**

1. Is the manuscript technically sound, and do the data support the conclusions?

Reviewer #1: Partly

Reviewer #2: Yes

2. Has the statistical analysis been performed appropriately and rigorously? 

Reviewer #1: Yes

Reviewer #2: Yes

3. Have the authors made all data underlying the findings in their manuscript fully available?

Reviewer #1: Yes

Reviewer #2: Yes

4. Is the manuscript presented in an intelligible fashion and written in standard English?

Reviewer #1: Yes

Reviewer #2: Yes

5. Review Comments to the Author

Reviewer #1: The manuscript PONE-D-22-19260 submitted by Mascheretti and colleagues in PlosONE has investigated pain and anxiety problems in adolescents regarding genetic factors and gene pathway-based analysis by performing genome-wide association study in two cohorts: the Quebec Newborn Twin Study (QNTS with 246 twin pairs and 321 parents) and the Longitudinal Study of Child Development in Quebec (QLSCD; with n=754 subjects).

Several suggestive genetic associations were observed for anxiety or pain scores but none reach the significant threshold for multiple tests. However, excess of association signals within the genes of the same biological process for both phenotypes were found. Several enriched pathways with nominally significant uncorrected p values were found for both cohorts. Several pathways/GO categories were shared between mean pain and anxiety score phenotypes.

The cohorts are well described, as well as, the scores for the anxiety and pain phenotypes. The genetic analysis with the genotyping, quality control and imputation are also clear, as well as, the pathway association analysis. The results described are in adequation with their observations. The introduction is short, the material and method section is clear, as well as, the results and discussion section.

I have a concern regarding the investigation of anxiety and pain in these cohorts. Evaluation of anxiety and pain between man and woman should be taking into account independently because the distribution of anxiety and pain is different between genders with an increase of anxiety and pain in girls than boys. Furthermore, this is also true and exacerbated for adolescents or young adult populations like the Quebec Newborn Twin Study and Longitudinal Study of Child Development in Quebec. How, the authors manage that? Why the authors did not investigate separately boys and girls?

Furthermore, for adolescent women, there is an increase of migraine and pain reported due to puberty and menstrual cycles. How the authors manage it for anxiety and pain phenotypes? This should also discuss in the paragraph of limitations.

Thus, it should be very important to have a specific investigation for girls only (and boy only) of this question of the genetic profile of anxiety and pain in adolescent.

Regarding the sample size of the two investigated cohorts, they are small. Thus, could you provide a computation of the statistical power of each of your cohort, and for the combination, for an expected significant genetic association with anxiety, pain and both? I guess this power is low and it will explain why the genome-wide association studies could not reach a significant threshold after multiple corrections.

Regarding Table 1, it is mentioned 41 pathways -page 8, line 183). However, in the table 1, there is only 37 GO set ID. Could you correct that sentence?

There is no figure of the genome-wide Manhattan plots for the two combined cohorts. This could be added in supplementary document.

Tables 1 to 3 are large and on several pages. Thus, I would move the table 3 in supplementary document.

In that state, the manuscript PONE-D-22-19260 submitted by Mascheretti and colleagues need some clarifications and answers to questions to be suitable for a publication in the journal PLOS One.

Reviewer #2: Thank you for the opportunity to review the manuscript entitled “Adolescent anxiety and pain problems: a joint, genome-wide investigation and pathway-based analysis”. Authors conducted a joint GWAS to get some meaningful findings. I have the following concerns.

1. It is well-known that anxiety and pain problems may be more prevalent in adult and old population than those in adolescent. However, this study used the adolescent population. Was the target population appropriate? Please state it.

2. It is crucial to identify MZ and DZ in twin study. How did the study identify the zygosity of the twin pairs?

3. How to avoid bias in the self-reported questionnaire of pain and anxiety to ensure the accuracy of the study?

4. Did the study consider the age, sex, etc. as the covariates in the data analysis?

6. PLOS authors have the option to publish the peer review history of their article (what does this mean?). If published, this will include your full peer review and any attached files.

Reviewer #1: No

Reviewer #2: **Yes: **Chunsheng Xu

---

## [Author Response · Author response to Decision Letter 0]

12 Apr 2023

Journal Requirements: 1. Please ensure that your manuscript meets PLOS ONE's style requirements, including those for file naming. The PLOS ONE style templates can be found at

A: We checked that our manuscript meets PLOS ONE’s style requirements.

JR: 2. Please provide additional details regarding participant consent. In the ethics statement in the Methods and online submission information, please ensure that you have specified what type you obtained (for instance, written or verbal, and if verbal, how it was documented and witnessed). If your study included minors, state whether you obtained consent from parents or guardians. If the need for consent was waived by the ethics committee, please include this information.

A: We provided additional details regarding participant consent in both the Methods (page 3, line 109 and page 4, lines 121-123) and online submission information.

JR: 3. PLOS requires an ORCID iD for the corresponding author in Editorial Manager on papers submitted after December 6th, 2016. Please ensure that you have an ORCID iD and that it is validated in Editorial Manager. To do this, go to ‘Update my Information’ (in the upper left-hand corner of the main menu), and click on the Fetch/Validate link next to the ORCID field. This will take you to the ORCID site and allow you to create a new iD or authenticate a pre-existing iD in Editorial Manager. Please see the following video for instructions on linking an ORCID iD to your Editorial Manager account: https://www.youtube.com/watch?v=_xcclfuvtxQ.

A: Thank you for this remark.

We updated the information.

JR: 4. Thank you for stating the following in the Acknowledgments Section of your manuscript:

“We are grateful to all families and participants who took part in the study. We thank the GRIP staff for data collection and management. Both studies were supported by grants from the Fonds de recherche du Quebec – Societe et Culture (FRQ-SC), Fonds de recherche du Quebec – Sante (FRQ-S), the Réseau québecois sur le suicide, les troubles de l’humeur et les troubles associés, the Social Science and Humanities Research Council of Canada (SSHRC), the Canadian Institutes for Health Research (CIHR), and Ste. Justine Hospital’s Research Center. The QNTS was also supported by funding from the National Health Research Development Program, Université Laval, and Université de Montreal. The QLSCD was also supported by funding from the Gouvernement du Québec, the Lucie and André Chagnon Foundation, the Robert-Sauvé Research Institute of Health and Security at Work, and the Institut de la statistique du Quebec. Dr Battaglia's research is supported by the Canadian Institutes of Health Research, the Natural Sciences and Engineering Research Council of Canada, the Quebec Pain Research Network, the Cundill Foundation and the CAMH Foundation; he has been supported by a research grant from the Université Laval Merck Sharpe Dome Foundation, has received stipends from the Canadian Psychiatric Association and from Servier International Medical Publishing Division. Dr. Boivin (Tier1), G. Rouleau (Tier 1), G. Turecki (Tier 1), I. Ouellet-Morin (Tier 2), J.-P. Gouin (Tier 2), and G. Garon-Carrier (Tier 2) are supported by the Canada Research Chair Program. Dr. Mascheretti was supported by Italian Ministry of Health Grants (Ricerca Corrente 2021). Dr. Fumagalli was supported by "5 per mille" funds for biomedical research”

“We are grateful to all families and participants who took part in the study. We thank the GRIP staff for data collection and management. Both studies were supported by grants from the Fonds de recherche du Quebec – Societe et Culture (FRQ-SC), Fonds de recherche du Quebec – Sante (FRQ-S), the Réseau québecois sur le suicide, les troubles de l’humeur et les troubles associés, the Social Science and Humanities Research Council of Canada (SSHRC), the Canadian Institutes for Health Research (CIHR), and Ste. Justine Hospital’s Research Center. The QNTS was also supported by funding from the National Health Research Development Program, Université Laval, and Université de Montreal. The QLSCD was also supported by funding from the Gouvernement du Québec, the Lucie and André Chagnon Foundation, the Robert-Sauvé Research Institute of Health and Security at Work, and the Institut de la statistique du Quebec. Dr Battaglia's research is supported by the Canadian Institutes of Health Research, the Natural Sciences and Engineering Research Council of Canada, the Quebec Pain Research Network, the Cundill Foundation and the CAMH Foundation; he has been supported by a research grant from the Université Laval Merck Sharpe Dome Foundation, has received stipends from the Canadian Psychiatric Association and from Servier International Medical Publishing Division. Dr. Boivin (Tier1), G. Rouleau (Tier 1), G. Turecki (Tier 1), I. Ouellet-Morin (Tier 2), J.-P. Gouin (Tier 2), and G. Garon-Carrier (Tier 2) are supported by the Canada Research Chair Program. Dr. Mascheretti was supported by Italian Ministry of Health Grants (Ricerca Corrente 2021). Dr. Fumagalli was supported by "5 per mille" funds for biomedical research.”

A: Thank you for this remark. 

Please update our Funding Statement as follows:

“Both studies were supported by grants from the Fonds de recherche du Quebec – Societe et Culture (FRQ-SC), Fonds de recherche du Quebec – Sante (FRQ-S), the Réseau québecois sur le suicide, les troubles de l’humeur et les troubles associés, the Social Science and Humanities Research Council of Canada (SSHRC), the Canadian Institutes for Health Research (CIHR), and Ste. Justine Hospital’s Research Center. The QNTS was also supported by funding from the National Health Research Development Program, Université Laval, and Université de Montreal. The QLSCD was also supported by funding from the Gouvernement du Québec, the Lucie and André Chagnon Foundation, the Robert-Sauvé Research Institute of Health and Security at Work, and the Institut de la statistique du Quebec. Dr Battaglia's research is supported by the Canadian Institutes of Health Research, the Natural Sciences and Engineering Research Council of Canada, the Quebec Pain Research Network, the Cundill Foundation and the CAMH Foundation; he has been supported by a research grant from the Université Laval Merck Sharpe Dome Foundation, has received stipends from the Canadian Psychiatric Association and from Servier International Medical Publishing Division. Dr. Boivin (Tier1), G. Rouleau (Tier 1), G. Turecki (Tier 1), I. Ouellet-Morin (Tier 2), J.-P. Gouin (Tier 2), and G. Garon-Carrier (Tier 2) are supported by the Canada Research Chair Program. Dr. Lampis was supported by Italian Ministry of Health Grants (Ricerca Corrente 2022, 2023). Dr. Fumagalli was supported by "5 per mille" funds for biomedical research.”

Review Comments to the Author

Reviewer #1: 

Reviewer: The manuscript PONE-D-22-19260 submitted by Mascheretti and colleagues in PlosONE has investigated pain and anxiety problems in adolescents regarding genetic factors and gene pathway-based analysis by performing genome-wide association study in two cohorts: the Quebec Newborn Twin Study (QNTS with 246 twin pairs and 321 parents) and the Longitudinal Study of Child Development in Quebec (QLSCD; with n=754 subjects).

Several suggestive genetic associations were observed for anxiety or pain scores but none reach the significant threshold for multiple tests. However, excess of association signals within the genes of the same biological process for both phenotypes were found. Several enriched pathways with nominally significant uncorrected p values were found for both cohorts. Several pathways/GO categories were shared between mean pain and anxiety score phenotypes.

The cohorts are well described, as well as, the scores for the anxiety and pain phenotypes. The genetic analysis with the genotyping, quality control and imputation are also clear, as well as, the pathway association analysis. The results described are in adequation with their observations. The introduction is short, the material and method section is clear, as well as, the results and discussion section.

Authors: We really thank Reviewer 1 who expressed favorable comments and provided useful suggestions.

R: I have a concern regarding the investigation of anxiety and pain in these cohorts. Evaluation of anxiety and pain between man and woman should be taking into account independently because the distribution of anxiety and pain is different between genders with an increase of anxiety and pain in girls than boys. Furthermore, this is also true and exacerbated for adolescents or young adult populations like the Quebec Newborn Twin Study and Longitudinal Study of Child Development in Quebec. How, the authors manage that? Why the authors did not investigate separately boys and girls?

A: Our previous work with the QNTS (PMID: 31944483) showed that, while the frequencies of endorsement vary according to sex (and quite in agreement with the Reviewer’s expectation), models that imposed no sex constraints had no worse fit than models that imposed such constraints. The implication is that, regardless of different prevalence in the two sexes, the etiology (i.e., same genes) can be assumed as comparable for boys and girls. This is now more explicitly explained in the introduction (page 2, lines 74-78):

“Similarly, the only two adolescent pain twin studies (one of which is longitudinal) [4] indicate that the co-occurrence of anxiety and adolescent pain problems is accounted for by genetic and environmental factors [4, 9] that are shared by the two phenotypes and has a similar etiology for boys and girls [4], even though both anxiety and pain problems are more prevalent among girls.”

(...), methods (page 7, lines 200-204):

“Multiple analyses of sex effects among QNTS participants showed that models that imposed sex-related constraints on anxiety or pain did not yield better fit than models that imposed no such constraints [4, 30]. Within the limitation of this quantitative genetic approach, this indicates a similar etiology for boys and girls alike for both anxiety and pain, and therefore participants of both sexes were pooled together in the analyses.”

(...), and discussion (limitations) sections (page 14, lines 367-369):

“Although available twin adolescent data indicate a similar etiology of pain and anxiety for boys and girls [4, 9, 30], these remain quantitative genetic modeling applications, so that molecular genetic replications in larger datasets are needed.”

We coherently added the following reference

[30] Battaglia M, Garon-Carrier G, Rappaport L, Brendgen M, Dionne G, Vitaro F, et al. Adolescent pain: appraisal of the construct and trajectory prediction-by-symptom between age 12 and 17 years in a Canadian twin birth cohort. Pain 2022; 163:e1013.

R: Furthermore, for adolescent women, there is an increase of migraine and pain reported due to puberty and menstrual cycles. How the authors manage it for anxiety and pain phenotypes? This should also discuss in the paragraph of limitations.

Thus, it should be very important to have a specific investigation for girls only (and boy only) of this question of the genetic profile of anxiety and pain in adolescent.

A: Our work with the QNTS (PMID: 34966130) showed that while the 6 pain items vary in frequency in boys and girls (and quite in agreement with the Reviewer’s expectation), adolescent pain best fits a unitary dimension rather than independent manifestations. Specifically, for frequent pain there was no symptom-by-sex interaction at any of the 5 longitudinal waves of assessment, which shows no significant variation between boys and girls for adolescent pain. Within the limits of the methods employed, this allows to use adolescent pain as a unitary score in the analyses.

This is now more explicitly explained in the methods (page 7, lines 200-204):

“Multiple analyses of sex effects among QNTS participants showed that models that imposed sex-related constraints on anxiety or pain did not yield better fit than models that imposed no such constraints [4, 30]. Within the limitation of this quantitative genetic approach, this indicates a similar etiology for boys and girls alike for both anxiety and pain, and therefore participants of both sexes were pooled together in the analyses.”

(...), and discussion (limitations) sections (page 14, lines 367-369):

“Although available twin adolescent data indicate a similar etiology of pain and anxiety for boys and girls [4, 9, 30], these remain quantitative genetic modeling applications, so that molecular genetic replications in larger datasets are needed.”

R: Regarding the sample size of the two investigated cohorts, they are small. Thus, could you provide a computation of the statistical power of each of your cohort, and for the combination, for an expected significant genetic association with anxiety, pain and both? I guess this power is low and it will explain why the genome-wide association studies could not reach a significant threshold after multiple corrections.

A: We conducted post-hoc power calculations to estimate the smallest genome-wide significant association that our samples could detect with 80% statistical power (α=5×10-8). Power analysis is now explicitly explained in the methods (page 6, lines 179-184 and page 7, lines 190-195):

“Post-hoc power calculations were conducted using the Genetic Power Calculator [28] to estimate the smallest effect size that our samples could detect with 80% statistical power. The analysis was modelled for an allele frequency of 0.05. No dominance effects and perfect linkage disequilibrium were assumed; alpha was set at 510-8. Under these assumptions, the minimal effect sizes predicted to be detectable with 80% power was 0.08% in our twin pairs’ cohort (n = 246). (…) As in the QNTS, post-hoc power calculations were conducted using the Genetic Power Calculator [28] to estimate the smallest effect size that our samples could detect with 80% statistical power. The analysis was modelled for an allele frequency of 0.05. No dominance effects and perfect linkage disequilibrium were assumed; alpha was set at 510-8. Under these assumptions, the minimal effect sizes predicted to be detectable with 80% power was 0.05% in our sample (n = 754).”

The following reference was coherently added:

[28] Purcell S, Cherny SS, Sham, PC. Genetic power Calculator: design of linkage and association genetic mapping studies of complex traits. Bioinformatics 2003; 19:149–150.

R: Regarding Table 1, it is mentioned 41 pathways -page 8, line 183). However, in the table 1, there is only 37 GO set ID. Could you correct that sentence?

A: We apologize for the mistake and thank you for spotting it. The correct number is 37. We have now modified the text accordingly (page 9, line 244).

By revising the manuscript, we realized that we made another mistake in reporting the results (cf. page 9, line 241). We coherently corrected this mistake in the revised version of the manuscript.

R: There is no figure of the genome-wide Manhattan plots for the two combined cohorts. This could be added in supplementary document.

A: Analysis of combined cohorts was performed by starting from clumped regions of the two datasets, not from individual SNPs. Specifically, we started from the top variants of both QNTS and QLSCD datasets, we generated two lists of genomic regions surrounding these variants and then, instead of analyzing them separately, we merged them and created a single region list. Given that, it is not possible to generate a SNP Manhattan plot for the combined region similar to those provided in the Supplementary Figures. We better specified the methodological procedure as follows (page 10, lines 282-283):

“(…) we merged the genomic regions for which we found nominally significant results in both the QNTS and QLSCD to achieve stronger statistical power and thus test the robustness of the previous findings. This led to a single region list containing the genomic regions surrounding the top variants of both QNTS and QLSCD samples.”

R: Tables 1 to 3 are large and on several pages. Thus, I would move the table 3 in supplementary document.

A: We agreed with the Reviewer Tables 1 to 3 are large and it would be better to remove them in supplementary materials. We accordingly renumbered previous supplementary tables.

R: In that state, the manuscript PONE-D-22-19260 submitted by Mascheretti and colleagues need some clarifications and answers to questions to be suitable for a publication in the journal PLOS One.

A: We believe we have a fine piece of work now and we hope to have provided all the requested clarifications and answers to be suitable for publication in PLoS One.

Reviewer #2: 

Reviewer: Thank you for the opportunity to review the manuscript entitled “Adolescent anxiety and pain problems: a joint, genome-wide investigation and pathway-based analysis”. Authors conducted a joint GWAS to get some meaningful findings. I have the following concerns.

Authors: We really thank Reviewer 2 who expressed favorable comments and provided useful suggestions.

R: 1. It is well-known that anxiety and pain problems may be more prevalent in adult and old population than those in adolescent. However, this study used the adolescent population. Was the target population appropriate? Please state it.

A: Pain problems not caused by identified medical conditions begins early in life, are common among children and adolescents (between 8% and 12% of adolescents aged 11–17 years and 16%-20% of youths or young adults; PMID: 20971561; PMID: 31954722; PMID: 18093737; PMID: 34966130; PMID: 31944483), are associated with internalizing symptoms (PMID: 31944483; PMID: 26901806; PMID: 28267063; PMID: 23940244), persist into youth and adulthood (PMID: 15109969; PMID: 28267063; PMID: 20971561; PMID: 18093737), and are associated with negative long-term outcomes (PMID: 31651579). Accordingly, using adolescent population to understand the nature of the co-occurrence between common and persistent pain and internalizing conditions (anxiety/depression), is necessary to inform diagnostic reasoning, guide clinical practice, develop new medications, and help reducing opioid prescriptions and abuse. Adolescent anxiety and persistent pain likely constitute a gateway to premature, more prolonged, and more hazardous opioid prescription (PMID: 30681713; PMID: 29532067; PMID: 27028915). Inasmuch as a culture of risk identification has not yet entered into practice, and specific treatments for young people who present with both pain and anxiety are lacking, this type of research can prove valuable. We addressed this issue in the “Conclusions” paragraph (pages 14-15, lines 386-389):

“Pain problems not caused by identified medical conditions begins early in life, are common among adolescents [1, 4, 8, 30, 57], are associated with internalizing symptoms [4, 6, 58-59], persist into youth and adulthood [1, 57-58, 60], and are associated with negative long-term outcomes [61].”

We coherently added the following references:

[57] Stanford EA, Chambers CT, Biesanz JC, Chen E. The frequency, trajectories and predictors of adolescent recurrent pain: a population-based approach. Pain 2008; 138:11-21.

[58] Rosenbloom BN, Rabbitts JA, Palermo TM. A developmental perspective on the impact of chronic pain in late adolescence and early adulthood: implications for assessment and intervention. Pain 2017; 158:1629-1632. 

[59] Shelby GD, Shirkey KC, Sherman AL, Beck JE, Haman K, Shears AR, et al. Functional abdominal pain in childhood and long-term vulnerability to anxiety disorders. Pediatrics 2013; 132:475-482. 

[60] Brattberg G. Do pain problems in young school children persist into early adulthood? A 13-year follow-up. Eur J Pain 2004; 8:187-199. 

[61] Murray CB, Groenewald CB, de la Vega R, Palermo TM. Long-term impact of adolescent chronic pain on young adult educational, vocational, and social outcomes. Pain 2020; 161:439-445.

R: 2. It is crucial to identify MZ and DZ in twin study. How did the study identify the zygosity of the twin pairs?

A: We thank the Reviewer for this remark.

We added details about the assessment of zygosity in the ‘Materials and Methods - Sample - Quebec Newborn Twin Study (QNTS)’ paragraph (page 3, lines 99-101):

“Zygosity was initially assessed via questionnaire [18] and confirmed with DNA tests on a subsample of same-sex pairs showing a 96% correspondence [19-20].”

Accordingly, we added the following references:

[18] Goldsmith HH. A zygosity questionnaire for young twins: A research note. Behavior Genetics 1991; 21:257–269.

[19] Forget-Dubois N, Perusse D, Turecki G, Girard A, Billette J-M, Rouleau G, et al. Diagnosing zygosity in infant twins: Physical similarity, genotyping, and chorionicity. Twin Research 2003; 6:479–485.

[20] Boivin M, Brendgen M, Dionne G, Ouellet-Morin I, Dubois L, Pérusse D, et al. The Quebec Newborn Twin Study at 21. Twin Research and Human Genetics 2019; 22:475–481.

R: 3. How to avoid bias in the self-reported questionnaire of pain and anxiety to ensure the accuracy of the study?

A: Previous studies using the same questionnaires implemented in this study reported acceptable level of internal consistency for the pain and anxiety scales. We have therefore revised the text as follows (page 4, lines 133-135 and page 5, lines 142-144): 

“Across the longitudinal assessment waves, the Cronbach’s alpha indices of the pain items’ cumulative scores (range: 0.71-0.74) and the anxiety symptoms mean scores (0.82-0.84) were acceptable [4]. (…) Average anxiety scores were then computed at each assessment time, with Cronbach alphas averaging 0.80 across measurements [23]”

We coherently added the following reference:

[23] Garmroudinezhad Rostami E, Touchette É, Huynh N, Montplaisir J, Tremblay RE, Battaglia M, et al. High separation anxiety trajectory in early childhood is a risk factor for sleep bruxism at age 7. Sleep 2020; 43:zsz317. 

R: 4. Did the study consider the age, sex, etc. as the covariates in the data analysis?

A: Age was not considered as covariate in our analysis because we addressed temporal stability of pain and anxiety symptoms by averaging scores across the different assessment waves. We addressed this issue in the Methods section (pages 4-5, lines 135-138 and page 5, lines 140-144)

“Because of the moderate-to-strong covariance of pain problems and anxiety symptoms across the 12, 13, and 14 assessment waves at age, suggesting temporal stability [4], we averaged these scores to obtain: a QNTS_Mean Pain, and a QNTS_Mean Anxiety index for each twin (S1 Table). (...) At age 12 and 13, each subject rated presence of headaches on a dichotomous (Yes/No) scale and 3 items for anxiety symptoms (i.e., “too fearful or anxious”, “worried” and “nervous, high-strung or tense”) on a 3-point scale ranging from 0 (never occurs) to 2 (frequently occurs). Average anxiety scores were then computed at each assessment time, with Cronbach alphas averaging 0.80 across measurements [23]”

Regarding sex, our previous work with the QNTS (PMID: 31944483) showed that, while the frequencies of endorsement vary according to sex (and quite in agreement with the Reviewer’s expectation), models that imposed no sex constraints had no worse fit than models that imposed such constraints. The implication is that, regardless of different prevalence in the two sexes, the etiology (i.e., same genes) can be assumed as comparable for boys and girls. This is now more explicitly explained in the introduction (page 2, lines 74-78):

“Similarly, the only two adolescent pain twin studies (one of which is longitudinal) [4] indicate that the co-occurrence of anxiety and adolescent pain problems is accounted for by genetic and environmental factors [4, 9] that are shared by the two phenotypes and has a similar etiology for boys and girls [4], even though both anxiety and pain problems are more prevalent among girls.”

(...), methods (page 7, lines 200-204):

“Multiple analyses of sex effects among QNTS participants showed that models that imposed sex-related constraints on anxiety or pain did not yield better fit than models that imposed no such constraints [4, 30]. Within the limitation of this quantitative genetic approach, this indicates a similar etiology for boys and girls alike for both anxiety and pain, and therefore participants of both sexes were pooled together in the analyses.”

(...), and discussion (limitations) sections (page 14, lines 367-369):

“Although available twin adolescent data indicate a similar etiology of pain and anxiety for boys and girls [4, 9, 30], these remain quantitative genetic modeling applications, so that molecular genetic replications in larger datasets are needed.”

We coherently added the following reference

[30] Battaglia M, Garon-Carrier G, Rappaport L, Brendgen M, Dionne G, Vitaro F, et al. Adolescent pain: appraisal of the construct and trajectory prediction-by-symptom between age 12 and 17 years in a Canadian twin birth cohort. Pain 2022; 163:e1013.

---

## [Editor Report · Decision Letter 1]

19 Apr 2023

Adolescent anxiety and pain problems: a joint, genome-wide investigation and pathway-based analysis

PONE-D-22-19260R1

Dear Dr. Battaglia,

We’re pleased to inform you that your manuscript has been judged scientifically suitable for publication and will be formally accepted for publication once it meets all outstanding technical requirements.

Kind regards,

Toryn Poolman

Academic Editor

PLOS ONE
---

## [Editor Report · Acceptance letter]

28 Apr 2023

PONE-D-22-19260R1 

Adolescent anxiety and pain problems: a joint, genome-wide investigation and pathway-based analysis 

Dear Dr. Battaglia:

I'm pleased to inform you that your manuscript has been deemed suitable for publication in PLOS ONE. Congratulations! Your manuscript is now with our production department. 

Kind regards, 

on behalf of

Dr. Toryn Poolman 

Academic Editor

PLOS ONE